# *Panicle Apical Abortion 7* Regulates Panicle Development in Rice (*Oryza sativa* L.)

**DOI:** 10.3390/ijms23169487

**Published:** 2022-08-22

**Authors:** Dongqing Dai, Huali Zhang, Lei He, Junyu Chen, Chengxing Du, Minmin Liang, Meng Zhang, Huimei Wang, Liangyong Ma

**Affiliations:** 1State Key Laboratory of Rice Biology and Chinese National Center for Rice Improvement, China National Rice Research Institute, Hangzhou 310006, China; 2Institute of Food Crops, Key Laboratory of Jiangsu Province for Agrobiology, Jiangsu Academy of Agricultural Sciences, Nanjing 210014, China

**Keywords:** rice, panicle apical abortion, 3′-UTR, LAX2

## Abstract

The number of grains per panicle significantly contributes to rice yield, but the regulatory mechanism remains largely unknown. Here, we reported a loss-of-function mutant, *panicle apical abortion 7* (*paa7*), which exhibited panicle abortion and degeneration of spikelets on the apical panicles during the late stage of young panicle development in rice. High accumulations of H_2_O_2_ in *paa7* caused programmed cell death (PCD) accompanied by nuclear DNA fragmentation in the apical spikelets. Map-based cloning revealed that the 3 bp “AGC” insertion and 4 bp “TCTC” deletion mutation of *paa7* were located in the 3′-UTR regions of *LOC_Os07g47330*, which was confirmed through complementary assays and overexpressed lines. Interestingly, *LOC_Os07g47330* is known as *FRIZZY PANICLE* (*FZP*). Thus, *PAA7* could be a novel allele of *FZP*. Moreover, the severe damage for panicle phenotype in *paa7/lax2* double mutant indicated that PAA7 could crosstalk with Lax Panicle 2 (LAX2). These findings suggest that PAA7 regulates the development of apical spikelets and interacts with LAX2 to regulate panicle development in rice.

## 1. Introduction

The abortion phenomenon of the panicle apical has a negative impact on grain yield, which acts as a severe problem in rice-breeding production. It is mainly induced by environmental and genetic factors. This phenomenon has been widely reported in rice, corn, millet, and other crops [1,2,3]. Many quantitative traits, such as *ds-1*~*ds-9*, *qMDS3*, *qDS9*, and *qPAA8* have been demonstrated to cause panicle apical abortion in rice [4,5,6]. Moreover, several functional genes responsible for panicle apical abortion have been identified, i.e., *TUT1* (*TUTOU1*), *OsALMT7* (*aluminum-activated malate transporter 7*), *OsCIPK31* (*calcineurin B-like protein-interacting protein kinase 31*), *SPL6* (*squamosa promoter binding protein-like 6*), *DPS1* (*degenerated panicle and partial sterility 1*), *PAA3* (*panicle apical abortion 3*), *CAX1a* (*Ca2+/H+ exchanger 1a*), *SUBSrP1* (*subtilisin-like Serine oprotease 1*), and *TUT2* [3,7,8,9,10,11,12,13]. Among them, mutation of *TUT1,* which encodes a SCAR/WAVE-like protein, shows defects in the arrangement of actin filaments in the trichome [3], while *OsALMT7* encodes an aluminum-activated malate transporter, preferentially expressed in the vascular tissues of developing panicles. The panicle apical portions in *OsALMT7* mutant *paab1-1* contains less malate than WT, and the injection of malate could alleviate the spikelet degeneration phenotype [13]. *DPS1* encodes a mitochondria-localized protein containing a cystathionine b-synthase domain, which plays an important role in regulating ROS homeostasis, anther cuticle formation, and panicle development in rice. DPS1 participates in ROS scavenging by interacting with Trx1 and Trx20 [9]. All the mutations of the above genes lead to unbalanced ROS homeostasis and programmed cell death (PCD) in spikelet cells of apical portions of panicle.

Grain weight, grain number per panicle, and panicle number are the main factors affecting rice yield [14]. Over the past four decades in China, the high production of rice benefits from the ever-increasing grain number per panicle [15], which is determined mainly by panicle branching. Therefore, exploring the molecular mechanisms underlying grain number per panicle is of great value for the improvement of rice yields. Currently, numerous genes that are involved in panicle branch formation have been identified. For instance, the decreased expression of *OsCKX2* (*Gn1a*), which encodes a cytokinin oxidase/dehydrogenase, promotes cytokinin accumulation in the inflorescence meristem and thereby increases the number of spikelets per panicle and rice yield [16]. Aside from genes responsible for enzymes, some transcriptional factors regulate panicle formation as well. The mutation of GRAS family transcription factor *MOC1*/*GNP6* shows defects in panicle branch formation and tiller number [17]. The primary function of *FRIZZY PANICLE* (*FZP*), encoding a transcription factor with ERF/AP2 domain, is to negatively regulate the production of axillary meristems and promote the establishment of floret meristems [18]. The 18 bp fragment insertion at 5.3 kb upstream of the *FZP* promoter inhibited *FZP* expression, resulting in an increase in number of secondary branches [19]. Overexpression of *MOTHER OF FT AND TFL1* (*OsMFT1*) can inhibit the transcription of *FZP*, thereby producing more secondary branches [20]. *LAX1* encodes a helix-loop-helix (bHLH) transcription factor, determines the initiation and maintenance of axillary meristem, and synergistically regulates sterile lemma development in rice with *FZP* [21]. The *lax1* mutant exhibits reduced tillers and lateral spikelets [22]. In addition, LAX1 interacts with OsPID and participates in the development of rice floral organs by regulating auxin transportation [23]. *LAX2*/*Gnp4* encodes a RAWUL domain protein and regulates the formation of axillary meristems [24]. A similar *lax1* phenotype of decreased secondary branches and spikelets per panicle was observed in *lax2* mutant [24]. LAX2 can interact with LAX1 to regulate the development of rice axillary meristems, and the *lax2*/*lax1* double mutant shows a more severe reduced-branching phenotype [25]. 

Here, we identified a new *FZP* allelic mutant *paa7* from a severe panicle mutant *abnormal panicle1* (*ap1*) and presented a novel function of *PAA7*, which controls panicle apical development. In addition, we further confirmed that PAA7 could interact with LAX2 to co-regulate the identification of the panicle branch. Our findings offer insights into the molecular mechanism underlying rice panicle formation and provide critical genetic resources for improving rice grain yields.

## 2. Results

### 2.1. Characterization of the paa7 Mutant

The *apical panicle abortion7* (*paa7*) mutant was obtained from *ap1*, a severe panicle mutant of an Egyptian rice variety “Giza159”, and showed a stable phenotype of panicle abortion. 

There was no significant difference in plant height and tiller number between wild- type (WT) Giza159 and *paa7* mutant (Figure 1A,B), but *paa7* mutant exhibited remarkably degenerated, whitish spikelets at the top of panicle after heading (Figure 1C). The degeneration appeared more frequently in the primary branches at the top of the panicle, indicating degradation degree was associated with spikelet location (Figure 1D,E). To verify this speculation, we performed statistics on degenerate spikelets of different branches. The results showed that, in the *paa7* mutant, approximately 20% of spikelets and 43% of primary branches, mainly located at the top of each panicle, were degenerated (Figure 1F). The degeneration length ranged from 0.6–7 cm, and the mean ratio of degeneration length per bleached primary branches was about 56% (Figure 1F). In addition, the panicle length, grain number per panicle, and 1000-grain weight of *paa7* mutant were lower than those of WT (Figure 1G–I).

To differentiate the abortion stage that emerged in *paa7*, we tracked the development of panicles from 1 cm in length. The results showed that the spikelets in the top panicle of *paa7* degenerated during the eighth stage of inflorescence development in the rice (In8) phenotype [26] when the panicle developed to 10–11 cm in length (Figure 1J). The aborted spikelets had no difference from normal spikelets before the 8 cm panicle, while they withered gradually after the In8 stage (Figure 1K). 

In addition to the aborted spikelets, other abnormal spikelets, such as ectopic lemma and palea, twisted glume, and shortened lemma, were observed in *paa7* (Appendix A). The floral organs, such as anthers, stigmas, and ovary, developed normally in *paa7*, even in degenerated spikelets (Appendix A). However, vascular bundle number in transverse sections of pedicels was reduced in *paa7* (Appendix A), indicating that material transport in the apical spikelet of *paa7* mutants could be impaired. Collectively, *paa7* is an apical panicle abortion mutant that significantly reduces the number of fertile spikelets and causes a massive loss in grain yield.

### 2.2. Enhanced Cell Death in paa7 Mutant

The cell death characteristic of degenerated spikelets in 7–8 cm panicle of WT and *paa7* mutant were identified through Trypan blue staining and Evans blue staining. The results showed that apical spikelets in *paa7* mutant were intensely stained but were hardly stained in WT (Figure 2A,B). 

The DNA damages in apical spikelets of 8 cm panicle were further evaluated through comet assay (single-cell gel electrophoresis assay), a simple and effective method for assessing DNA damage in cells [27], and the results indicated significantly increased DNA damage in top spikelets of *paa7* compared with WT (Figure 2C–F). In addition, apoptosis in the spikelets of the 8 cm panicle was detected through TUNEL (terminal deoxynucleotidyl transferase-mediated dUTP nick-end labeling) assay. Compared with WT, the TUNEL-positive signals were much stronger in *paa7* (Figure 2G,H). These results suggested that DNA fragmentation occurred at a single-cell level and therewith caused cell death in spikelets and anthers in *paa7* mutant.

Excessive ROS accumulation can induce cell apoptosis through the oxidative stress response. Hydrogen peroxide is a signal substance that causes cell death. The cellular ROS levels were monitored with 3,3′-diaminobenzidine (DAB), and the results showed that spikelets of *paa7* can be dyed brown, especially in the top spikelets. In contrast, spikelets of WT were hardly stained (Figure 3A), indicating increased ROS levels in degenerated spikelets. The concentration of H_2_O_2_ and malondialdehyde (MDA) were significantly increased in apical spikelets of *paa7* over those of WT after 8–10 cm panicle in length (Figure 3B–D). All these results indicated that the over-accumulation of oxidant products caused cell death and eventually led to panicle apical abortion in *paa7*.

### 2.3. Mutations in 3′-UTR of PAA7 Responsible for Panicle Apical Abortion 

To reveal the genetic features of *paa7*, we developed two populations derived from the cross of *pa**a7* and 9311, II-32B, respectively. All F_1_ progenies demonstrated a similar wild-type panicle phenotype. In F_2_ population, the separation ratio of the phenotype of wild-type to panicle apical abortion corresponded to the separation ratio of 3:1 (Appendix A), indicating that the panicle apical abortion trait of *paa7* was controlled by a recessive nuclear gene.

The *PAA7* gene was initially located between RM1209 and RM1357 on the long arm of chromosome 7 with 284 SSR markers using 349 recessive individuals from the F_2_ mapping population of *pa**a7* and 9311 (Figure 4A). Subsequently, we delimited *PAA7* to a 28 kb interval between the markers I12 and RM22120 using 1580 F_2_ recessive plants in 2018 (Figure 4B,C). 

There were six predicted coding genes located in this interval region (http://www.gramene.org/, accessed on 1 July 2021) (Figure 4C). To further determine the candidate gene of *PAA7*, the entire sequences of the six genes were sequenced. Sequence comparison between *paa7* and WT revealed that a nucleotide substitution A-G and T-C was found in exon of *LOC_Os07g47320* and *LOC_Os07g47340*, respectively. Further, a 3 bp (AGC) insertion and a 4 bp (TCTC) deletion were found in the 3’UTR of *LOC_Os07g47330* (*FZP*) (Figure 4D,E). No other nucleotide variations were found in the other three genes. 

To verify the putative *PAA7 gene*, *LOC_Os07g47320, LOC_Os07g47330* and *LOC_Os07g47340* complementation assays were performed in *paa7*, respectively. The results showed that only transformants with *LOC_Os07g47330* complementary vector rescued the *paa7* phenotype (Figure 4F,G). Transgenic plants with overexpressed *LOC_Os07g47330* also displayed the normal panicle phenotype as expected (Figure 4H). Expression analysis revealed that the expression level of *LOC_Os07g47330* was about 20% decreased in *paa7* compared with wild-type in 7–8 cm young panicle (Figure 4I). 

Taken together, these results suggested that the candidate gene of *PAA7* was *LOC_Os07g47330*, which shared the common locus with the previously reported *FZP* [18], and the mutations in 3′-UTR of *PAA7* were responsible for panicle apical abortion. 

### 2.4. Feeding of Exogenous Auxin Alleviates Apical Spikelet Degeneration in paa7

As the mode of panicle abortion in *paa7* is consistent with spatiotemporal features of apical dominance mediated by auxin, we hypothesized that the degenerate phenotype of the top spikelet in *paa7* might be related to auxin regulation. The endogenous indole-3-acetic acid (IAA/auxin) concentration was measured in WT and *paa7* at different panicle development stages. The results showed that the auxin content of *paa7* was slightly lower during the early stage of panicle development compared with wild type; however, it significantly decreased after the panicle developed to 8 cm in length (Figure 5A).

To confirm that decreased auxin might be responsible for apical panicle abortion, 10^−6^–10^−3^ M concentration of exogenous auxin were sprayed every day since the In1 stage (Figure 5B–F). We found that the degenerate phenotype occurred in panicles of WT plants with all concentrations of IAA treatment, and the degeneration was positively correlated with IAA concentration. However, the apical spikelet abortion phenotype in *paa7* was alleviated under 10^−6^ M and 10^−5^ M exogenous auxin treatment (Figure 5B–D,G–I), while more severe panicle apical abortion was displayed under 10^−4^ M and 10^−3^ M exogenous auxin treatment (Figure 5E–I). These results suggested that insufficient synthesis of auxin may be responsible for the panicle apical abortion phenotype in *paa7*. 

### 2.5. PAA7 Activates the Expression of OsYUCCA6 to Regulate the Auxin Content in Young Panicle

In order to detect the effect of PAA7 on the regulation of auxin-, the expression level of auxin synthesis-associated genes was detected in apical spikelets tissue of panicle 6 cm–8 cm in length. A total of 22 genes related to auxin synthesis were selected for RT-qPCR analysis. The results showed that the expression level of 9 genes, including *OsYUCCA2*, *OsYUCCA5*, *OsYUCCA6*, *OsSPL10*, *HL6*, *OSH1*, *OsHOX1*, *OsHOX28*, and *OsFTIP7*, were significantly changed in *paa7* (Appendix A). 

Yeast one-hybrid assay was further carried out to verify the interaction between PAA7 and the promoter of these nine genes. The results indicated that only the promoter regions of *OsYUCCA6* could interact with PAA7 in yeast (Figure 5J). Chromatin immunoprecipitation (ChIP)-qPCR showed that PAA7 was significantly enriched in *OsYUCCA6-7* promoter regions (Figure 5K) in vivo, which contained MYB recognition site (CCGTTG) through *cis*-element analysis. In addition, EMSA analysis was performed to verify that PAA7 can bind directly to the 59-bp fragment containing the MYB recognition site of *OsYUCCA6* in vitro (Figure 5L). Taken together, PAA7 activates auxin biosynthesis by directly binding to the MYB recognition site in the promoter region of *OsYUCCA6*.

### 2.6. PAA7 Synergistically Works with LAX2 during Panicle Development

Both *paa7* (Figure 6A) and previously reported *lax2-4* (Figure 6B) mutants [28] were derived from a same natural panicle mutant *ap1* (Figure 6C–E), which shows frizzy panicle phenotype and is similar to severe *fzp* allelic mutant. We speculate that *LAX2-4* and *PAA7* may synergistically regulate rice panicle development. To further elucidate this speculation, reciprocal crosses were performed between *lax2-4* and *paa7*. All F_1_ progenies showed a wild-type panicle phenotype. However, in F_2_ population, the separation ratio of the phenotype of wild-type, panicle apical abortion (*paa7)*, lax panicle (*lax2-4*) and frizzy panicle (*ap1*) basically corresponded to the segregation ratio of 9:3:3:1 (Appendix A). Interestingly, the phenotype of the original *ap1* mutant, which exhibited curled panicle branches and almost no spikelets (Figure 6C,D), was separated in F_2_ population. The spikelets were replaced by higher-order vermiculate branches in *lax2-4*/*paa7* double mutant (Figure 6E), similarly with *fzp* strong mutant’s phenotype reported previously [18]. These results suggested a possible genetic interaction existed between *LAX2-4* and *PAA7*.

In addition, a CRISPR/Cas9-LAX2 knockout vector was introduced into the *paa7* mutant and four T_1_ independent homozygous lines were obtained (Figure 6F). All these four lines showed a similar panicle phenotype with *ap1*
Figure 6G–J and Appendix A), indicated that the mutation of *LAX2* will aggravate *paa7* phenotype. Thus, we speculated that *LAX2* and *PAA7* work synergistically in regulating the transition from branch meristem to spikelet meristem during panicle development. 

To explore the relationship between LAX2 and PAA7, yeast two-hybrid assay was performed and the results showed that LAX2 could interact with PAA7 in yeast (Figure 7A). Further pull-down, co-immunoprecipitation and bimolecular fluorescent complimentary assay demonstrated that LAX2 could directly interact with PAA7 (Figure 7B–D). These results suggested that PAA7 interacts with LAX2 to co-regulate panicle development.

## 3. Discussion

### 3.1. paa7 Is a Novel Allele of fzp

*FZP* encodes an AP2 family transcription factor and plays essential roles in rice panicle development [18]. To date, 18 *fzp* mutants have been reported in rice [18,19,21,29,30,31], with most due to mutation in the coding or 5’UTR region of *FZP*, causing multiple phenotypes [31,32]. Weak *fzp* mutants like *sgdp7* or *qSrn7* have smaller but more spikelets per panicle [19,30]; *fzp-12* and *fzp-13* display defects in the sterile lemma identity [29]. However, severe *fzp* mutants like *fzp-1* show a frizzy panicle phenotype, similar to stick panicles, and the spikelets are replaced with masses of higher-order branches rather than glumes and florets [18]. 

In this study, we characterized a new allelic mutant of *FZP, paa7,* displaying typical panicle apical abortion (Figure 1C), which has never been reported in previous *FZP* allelic mutants. The variations located in 3′-UTR of *PAA7* were responsible for the phenotype of panicle apical abortion in *paa7* (Figure 4); 3′-UTR is unique in eukaryotes, which contains various cis-acting elements and involves post-transcriptional regulation by binding with trans-acting factors or microRNAs [33]. In this study, the expression level of *PAA7* was reduced by 20% in *paa7* mutant because of the 3 bp “AGC” insertion and 4 bp “TCTC” deletion in 3′-UTR of *PAA7* (Figure 4I). The complementary assays and overexpressed lines confirmed that the mutations in 3′-UTR of *PAA7* were responsible for panicle apical abortion (Figure 4F–H). 

Collectively, *paa7* is a novel allele of *fzp*, with a new phenotype of panicle apical abortion and new functional variations in 3′-UTR.

### 3.2. Panicle Abortion in paa7 Possibly Be Caused by Down-Rugulation of OsYUCCA6

Panicle apical abortion in rice can lead to a decrease in grain yield. To date, nine panicle apical abortion genes have been identified. These genes are involved in many biological processes, such as maleic acid transport (*ALMT7*), endoplasmic reticulum stress (*SPL6*), and redox balance (*DPS1*) [8,9,13], etc. However, it remains unclear whether auxin synthesis accounts for panicle apical abortion.

In this study, the development of apical spikelets in *paa7* was hindered, which may be ascribed to the impaired material transportation in the panicle. Compared with WT, the content of IAA/auxin of the apical panicles, the vascular bundles’ number in the pedicel’s transverse section, and the number of sclerenchyma cells were reduced significantly in *paa7* mutant (Appendix A), which caused weakened branch transport capacity. We hypothesized that auxin deficiency leads to failure of vascular bundle development and further collapse of material transport and accumulation of intracellular ROS. The content of endogenous auxin and response to appropriate concentration (10^−5^–10^−6^) of exogenous auxin treatment in *paa7* confirmed this speculation (Figure 5B–D). However, excessive auxin (10^−3^–10^−4^) had opposing effects, in that it aggravated the degeneration phenotype in *paa7* (Figure 5E,F), which corresponds to previous studies [7]. Thus, the panicle apical abortion of *paa7* was associated with auxin.

*YUCCAs*, which are involved in auxin synthesis through the Trp-dependent pathway, can convert tryptamine (TAM) to N-hydroxytryptamine (NHT) in vitro [34]. *OsYUCCA6* is preferentially expressed in the top of coleoptiles and is involved in IAA biosynthesis in rice [35]. The decreased expression levels of *OsYUCCA6* may lead to lower auxin content in young panicles of *paa7* mutant. We speculate that the down-regulation of *OsYUCCA6* led to auxin insufficient in young panicles of *paa7*. However, why auxin deficiency leads to apical abortion in rice still needs to be further explored.

### 3.3. LAX2 and PAA7 Interact to Regulate Panicle Development in Rice

*LAX2-4*/*LAX2* and *PAA7*/*FZP* play important roles in rice panicle development and are involved in the regulation of the transition from branch meristem to spikelet meristem with opposite effects [24,31]. Previous research suggests that both *LAX2* and *FZP* could interact with *LAX1*, which was also known in rice panicle and floret development [21,36]. Tillers, secondary branches, and lateral spikelets were incapable of development in *lax1*/*lax2* double mutants [36], while the branches developed normally but failed to produce spikelets in *lax1*/*fzp* double mutant. While LAX1 and FZP could synergistically control sterile lemma development [21], the interaction between *LAX2* and *FZP* remains unclear.

As mentioned above, both *paa7* mutant and *lax2-4* mutant were derived from the same panicle defective *ap1* mutant. The phenotype of *paa7*/*lax2-4* double mutant was consistent with *ap1* mutant (Figure 6C–E), and the loss of functional *LAX2-4* aggravated the panicle defect of *paa7* (Figure 6F–J). In *lax2-4*/*paa7* double mutants, branch meristems could not be transformed into spikelet meristems normally, but reversely showed the characteristics of axillary meristem and continuously produced new high-order branches (Figure 6C–E). The protein interaction experiments also verified the interaction between LAX2-4 and PAA7/FZP (Figure 7). Thus, LAX2-4 and PAA7/FZP work synergistically to regulate the differentiation and meristem identity transformation of branch meristems during young panicle development. 

One model was proposed to explain the interaction between LAX2 and PAA7/FZP (Figure 8). In WT, *PAA7*/*FZP* and its related genes are expressed normally to produce healthy panicles. However, the expression of *PAA7*/*FZP* and its downstream auxin-related genes are suppressed and finally caused panicle apical abortion in *paa7*. In addition, the adverse panicle of *paa7* was enhanced by the loss of functional LAX2 (Figure 8).

## 4. Materials and Methods

### 4.1. Plant Materials and Growth Condition

The *paa7* mutant was obtained from a severe panicle mutant of an Egyptian rice variety, “Giza159”, after trait isolation and phenotypic stabilization. 

Rice plants were grown in the paddy field at China National Rice Research Institute (CNRRI) regularly. The transgenic plants were grown in the net house with a 14 h light (30 °C)/10 h dark (28 °C) light cycle at CNRRI. All plants were grown following standard agricultural field management.

### 4.2. Histological Analysis and TUNEL Assay

For histological observation, paraffin section was performed. Briefly, fresh spikelets and branches from wild type and *paa7* were placed in 50% formalin–acetic-acid–alcohol mixed solution and treated as previously described [37]. Then, the paraffin sections were stained with toluidine blue and observed through microscope (Nikon ECLIPSE E100 Nikon Imaging (Shanghai, China) Sales Co., Ltd.).

For TUNEL assay, apical spikelets of 7–8 cm panicle in length of WT and *paa7* were fixed in the FAA fixation solution overnight at 4 °C. The subsequent steps were taken according to the previously described method [13].

### 4.3. Trypan Blue and Evans Blue Staining

For trypan blue staining, apical spikelets of 7–8 cm panicle in length of WT and *paa7* were incubated with trypan blue solution (0.81% NaCl, 0.06% KH_2_PO_4_ and 0.4% trypan blue) for 10 min at 100 °C. After incubation, 2.5 mg/mL of chloral hydrate solution was used to decolorize. 

For Evans blue staining, apical spikelets of 8 cm panicle in length of WT and *paa7* were incubated with Evans blue solution (0.25%, *w*/*v*) overnight at room temperature. After incubation, absolute alcohol was used to decolorize. Photographs were taken with a digital camera (D810, Nikon Imaging (Shanghai, China) Sales Co., Ltd.).

### 4.4. Comet Assay

The comet assay was performed with the Single Cell Gel Electrophoresis Assay Kit (Trevigen, Gaithersburg, MD, USA) according to the manufacturer’s instructions. The comet nuclei were visualized by confocal microscopy (LSM 780; Zeiss) at 480 nm; 30 comet nuclei in each slide were analyzed by Comet Assay Software Project (CASP, http://www.casp.of.pl/). 

### 4.5. Determination of H_2_O_2_ and Malondialdehyde (MDA) Contents

The H_2_O_2_ and MDA contents were determined with spikelets of 7–8 cm panicle using H_2_O_2_ content assay kit (Comin, Suzhou, China) and MDA content assay kit (Comin, China), respectively, according to the manufacturer’s instructions.

For 3,3-diaminobenzidine (DAB) staining, after incubation in 1 mg/mL DAB solution overnight at room temperature and decolorized with absolute alcohol, spikelets of WT and *paa7* samples were observed and photographed with a digital camera (D810, Nikon). 

### 4.6. Positional Cloning of the PAA7 Gene

Two F_2_ populations derived from the cross between *paa7* mutant and indica cultivar 9311, II-32B were constructed for genetic analysis and physical mapping. 

F_2_ plants with recessive *paa7* phenotype were used for genetic mapping, while 46 InDel and SNP markers were developed by comparing the genomic sequences of Nipponbare and 9311 between the markers RM1209 and RM1307 for fine mapping. 

All the markers used in this study are listed in Appendix A.

### 4.7. RNA Extraction and Quantitative Real-Time PCR Analysis

Total RNA was extracted from young panicle tissues using the Total RNA Miniprep kit (Axygene, Shanghai, China); cDNA was synthesized with 1 μg RNA using the ReverTra Ace qPCR-RT kit (Toyobo (Shanghai, China) BIOTECH Co., Ltd.). RT-qPCR was performed using PowerUp™ SYBR™ Green mix (Thermo Fisher Scientific, Waltham, MA, USA) in an Applied Biosystems 7900HT instrument. Rice *Ubiquitin* (*LOC_Os03g13170*) was used as an endogenous control. The primers used for RT-qPCR are listed in Appendix A. Data were analyzed following the relative quantification method [38].

### 4.8. Vector Construction and Transformation

For the complementary assay, full-length gDNA of *LOC_Os07g47320, LOC_Os07g47330*, and *LOC_Os07g47340* were cloned into *pCAMBIA1300 BamH* I site using ClonExpress II One Step Cloning Kit (Vazyme, Nanjing, China), respectively. For the over-expression vector, full-length cDNA of *LOC_Os07g47330* was cloned into *pUBI-Flag-GFP Kpn* I site under the control of *Maize* Ubiquitin promoter. CRISPR/Cas9 vectors were constructed according to a previous report [39]. 

All the vectors were transformed by *A. tumefaciens*-mediated transformation as previously described [40].

### 4.9. Yeast Two-Hybrid (Y2H) Assay

The CDS of *LAX2* and *PAA7* amplified from Nipponbare cDNA were cloned into the pGBKT7 or pGADT7 vector, respectively. Yeast two-hybrid analysis was performed with Matchmaker Gold Yeast Two-Hybrid System (Clontech, Beijing, China) according to the manufacturer’s instructions. The primers used are listed in Appendix A.

### 4.10. Pull-Down Assay

The purified recombinant protein was incubated in pull-down buffer (50 mM Tris-HCl (pH 7.5), 5% glycerol, 1 mM EDTA, 1 mM DTT, 1 mM PMSF, 0.01% Nonidet P-40, 150 mM KCl) for 2 h at 4 °C with 20 μL His-beads per tube. After incubation, beads were washed 6 times with pull-down buffer. The sample was denatured at 100 °C for 5 min in 50 μL SDS loading buffer and analyzed by SDS-PAGE; anti-MBP and anti-His were used to detect the samples, respectively.

### 4.11. Bimolecular Fluorescence Complementation (BiFC) Assay

For BiFC assays, full-length CDS of *LAX2* and *PAA7* were cloned into1300-YN vector or 2300-YC vector, respectively. The constructs were co-expressed in rice protoplasts. YFP signals were observed about 24 h after expression. The primers used are listed in Appendix A.

### 4.12. Coimmunoprecipitation (Co-IP) Assay

Both the *p35S-PAA7-GFP* and *p35S-LAX2-HA* vectors were co-expressed in rice protoplasts for 24 h, and nucleoprotein was extracted with protein extraction buffer. Then, 25 µL of anti-HA beads were added after equilibrium to the extracted protein and incubated at room temperature for 0.5 h. After incubation, the beads were washed five times with 5X TBST buffer; then, 100 µL 50 mM NaOH was added, and the samples were boiled for 5 min and centrifuged. The supernatant was separated by 10% SDS–PAGE. Anti-GFP antibody (Abmart, M20008; 1:5000) and anti-HA antibody (Abmart, M20008; 1:5000) were used for further study.

### 4.13. Auxin Content Determination

Determination of IAA content was carried out using the enzyme-linked immunosorbent assay (ELISA) method as previously described [41]. Briefly, 0.1 g panicle tissue was ground to a fine powder in liquid nitrogen and homogenized with 900 μL PBS buffer (PH 7.2–7.4). After centrifuging at 3000× *g* for 20 min at 4 °C, the supernatant was used to determine the content of auxin. The subsequent steps follow the manufacturer’s instructions of plant indole acetic acid (IAA) ELISA kit (Dogesce, Beijing, China).

### 4.14. Yeast One-Hybrid Assay

The promoter sequence (1000 bp upstream of the transcriptional start site) of *OsYUCCA2*, *OsYUCCA5*, *OsYUCCA6*, *OsSPL10*, *HL6*, *OSH1*, *OsHOX1*, *OsHOX28* and *OsFTIP7* was amplified and cloned in the pLacZi vector. The *PAA7* full-length cDNA sequence was inserted into the pB42AD vector. Plasmids were transformed into yeast strain EGY48. Transformed yeast strains were grown on SD/-Ura/-Trp plates. The interaction was confirmed by blue colonies on the SD/-Trp/-Ura medium containing X-gal.

### 4.15. Chromatin Immunoprecipitation-Quantitative PCR (ChIP-qPCR)

Chromatin was isolated from 2–4 g leaves of OE-*FZP* plants according to the previous method [42]. The following steps were performed as previously described [42,43]. Gene-specific primers were selected in the 1500 bp promoter region of *OsYUCCA6*. The input and immunoprecipitated DNA were used for qRT-PCR with gene-specific primers, respectively. Each sample was repeated three times. The qPCR results were analyzed according to the manual of the Magna ChIP HiSens kit (Millipore).

### 4.16. Electrophoresis Mobility Shift Assay (EMSA)

Full-length CDS of *PAA7* were cloned into the *pMAL-C2X* vector by fusing the MBP tag to their N termini. The MBP-PAA7 fusion protein was purified with the PurKine™ MBP-Tag Protein Purification Kit (Dextrin) (Abbkine, USA). The EMSA probes in a length of 59nt in the *OsYUCCA6* promoter were synthesized and labeled with Cy5.5 by Tsingke Biotechnology Co., Ltd. (Beijing, China). The reaction was performed as described by [43]. The fluorescence signal was visualized using an Odyssey CLx infrared fluorescence imaging system (LI-COR). The primers used in this experiment are listed in Appendix A.

## 5. Conclusions

In this study, we reported a panicle apical abortion mutant *paa7*. The mutation in the 3’-UTR regions of *LOC_Os07g47330* disrupted the normal transcript of *PAA7* and caused panicle apical abortion in rice. Moreover, PAA7 could crosstalk with LAX2 to co-regulate the panicle development in rice. In summary, our results emphasize the importance of *PAA7* in the young panicle regulation network and provide insights into the mechanism of panicle and spikelet development in rice.

## Figures and Tables

**Figure 1 ijms-23-09487-f001:**
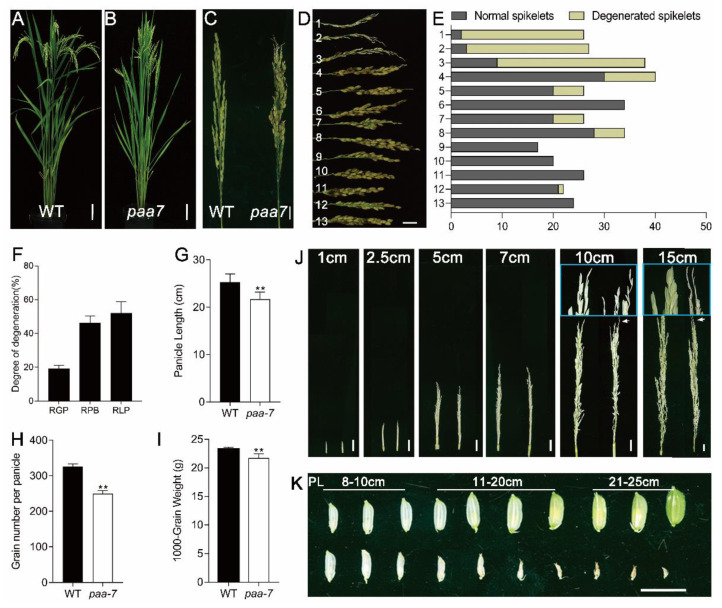
Phenotypic characterization of *paa7*. (**A**,**B**) Morphology of WT and *paa7* plants, bars = 10 cm. (**C**) Panicle of WT and *paa7*, bars = 2 cm. (**D**) Phenotype of primary branches from one panicle; 1–13 indicate primary branches from the top to the bottom of the panicle in *paa7*, bar = 1 cm. (**E**) Statistics for the spikelet number correspond to (**D**). (**F**–**H**) Quantification of panicle length, grain number per panicle, 1000-grain weight. (**I**) Degree of degeneration of *paa*; *7*RGP: ratio of degenerated grains per panicle (%), RPB: ratio of degenerated primary branches (%), RLP: ratio of degenerated length per primary branch (%). Error bars indicate standard error. ** *p*  ≤ 0.01 by Student’s *t*-test, *n* = 10. (**J**) Representative images of WT (left) and *paa7* (right) developing panicles; insets show magnified views of the top of panicles, bars = 1 cm. (**K**) Spikelets of panicles after 8 cm of WT and *paa7*; the number indicates the length of the panicle from which they were collected, bar = 1 cm.

**Figure 2 ijms-23-09487-f002:**
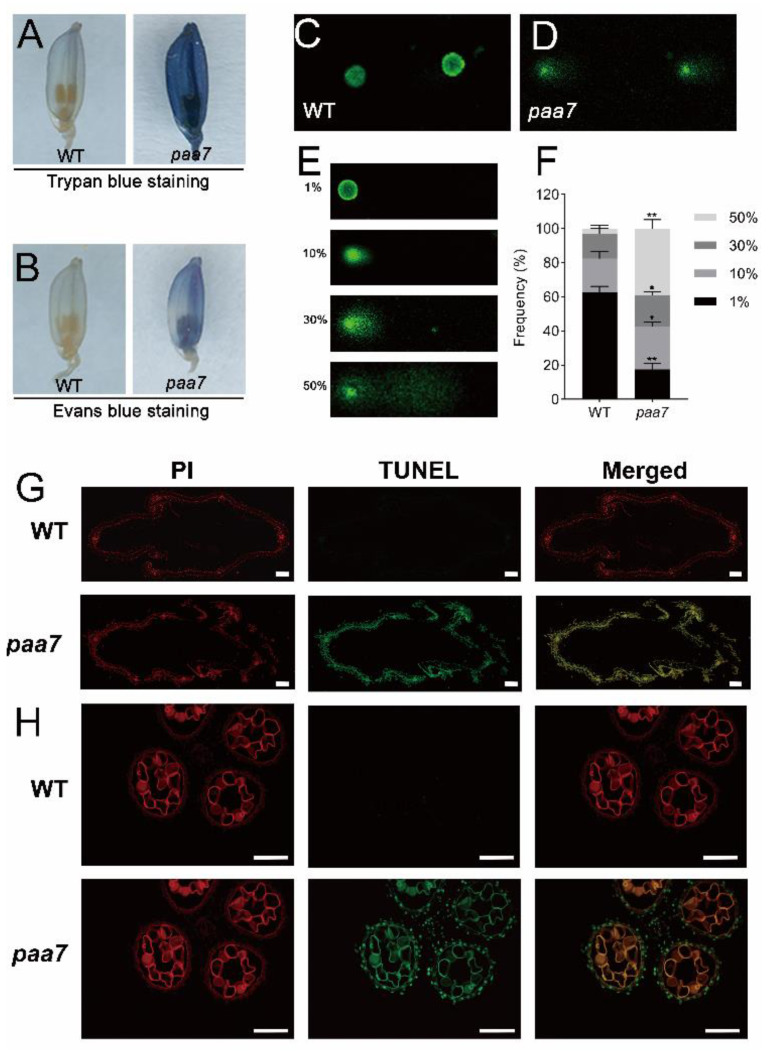
Cell-death-related events are triggered in *paa7*. (**A**,**B**) Trypan blue and Evans blue staining of apical spikelets. (**C**,**D**) Typical comets for DNA damage in nuclei of the apical spikelet cells. (**E**) Four types of DNA damage extents in each nucleus are indicated by the units 1%, 10%, 30%, or 50%. An increased unit and anthers of WT and *paa7* correlated with a larger comet tail. Bars = 100 μm. (**F**) Frequency distribution of four types of DNA damage extents in WT and *paa7*, data are means ± SD (*n* = 3). Asterisks indicate significant difference by Student’s *t*-test (** *p* < 0.01 and * *p* < 0.05). (**G**,**H**) TUNEL analysis of the apical spikelet cells of glumes, Bars = 100 μm.

**Figure 3 ijms-23-09487-f003:**
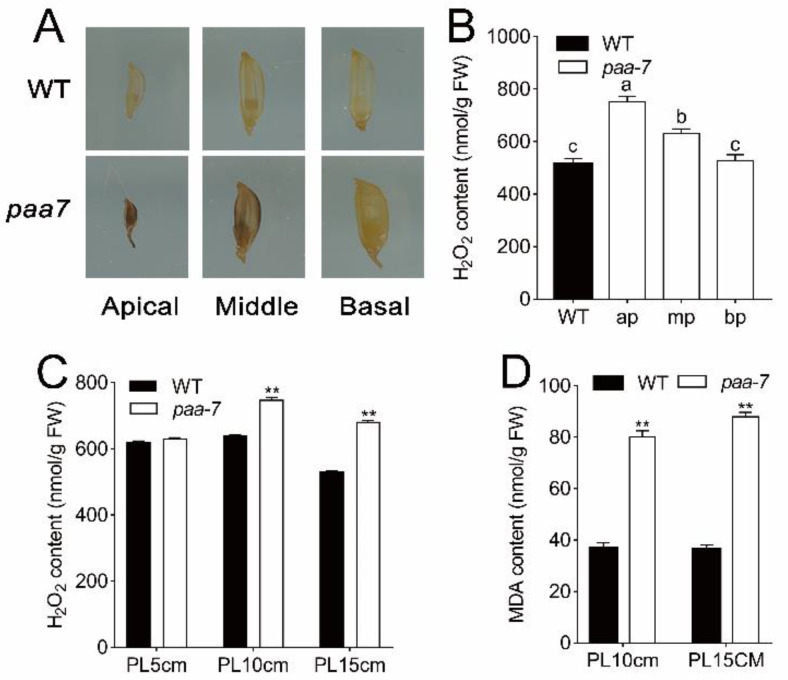
Determination of ROS content in WT and *paa7*. (**A**) DAB staining of spikelets in different parts of rice panicle. (**B**) H_2_O_2_ content in spikelets of apical part of the panicle in WT, apical part (ap), middle part (mp), and bottom part (bp) of panicle in *paa7*; data are means ± SD (*n* = 3); different letters indicate significant differences by one-way ANOVA and Duncan’s test (*p* < 0.05). (**C**) H_2_O_2_ content in apical spikelets at different stages of panicle development. (**D**) MDA content in apical spikelets at different stages of panicle development. Data are means ± SD (*n* =3); asterisks indicate significant differences by Student’s *t*-test (** *p* < 0.01).

**Figure 4 ijms-23-09487-f004:**
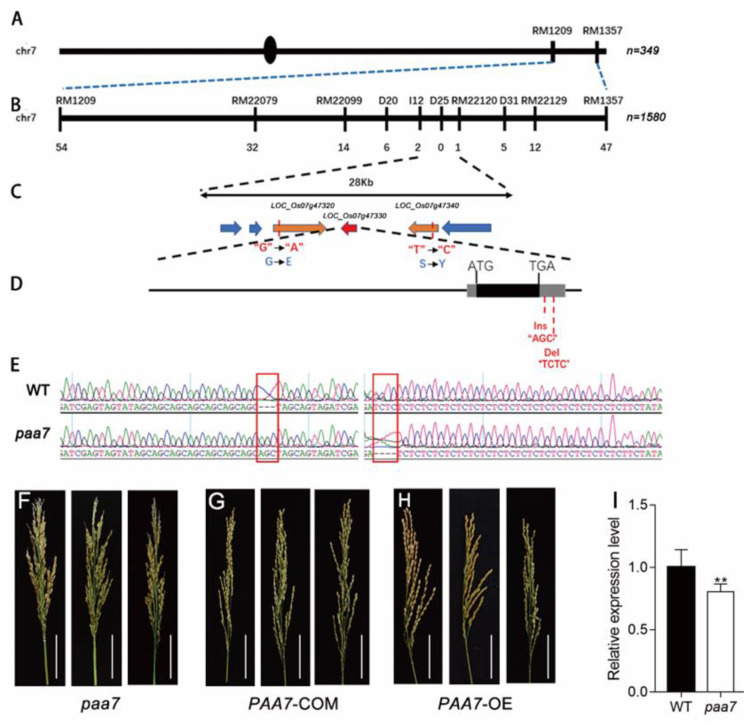
Mutation of 3′-UTR of *PAA7* caused panicle apical abortion. (**A**) The *PAA7* locus was mapped to the long arm of chromosome 7 between the markers RM1209 and RM1357. (**B**) Mapping of the *PAA7* locus between markers I12 and RM22120. ‘n’ represents homozygous recessive individuals derived from an F_2_ population from the cross *paa7*  ×  II-32B, *paa7* × 9311. The number of recombinants is indicated below the map. (**C**) The *PAA7* locus was narrowed down to a 28 kb region, including six putative protein-encoding genes. (**D**) There are 3 bp (AGC) insertion and 4 bp (CTCT) deletion in the 3′UTR in *FZP* of *paa7* mutant. (**E**) Sequencing results of WT and *paa7* gDNA, red box indicated the mutant site. (**F**–**H**) Panicles of *paa7* mutant plants (**F**), complementary transgenic plants (**G**) and overexpressed transgenic plants (**H**), Bars = 5 cm. (**I**) Expression of *FZP* in WT and *paa7* mutant, data are means ± SD (*n* =3), asterisks indicate significant difference by Student’s *t*-test (** *p* < 0.01).

**Figure 5 ijms-23-09487-f005:**
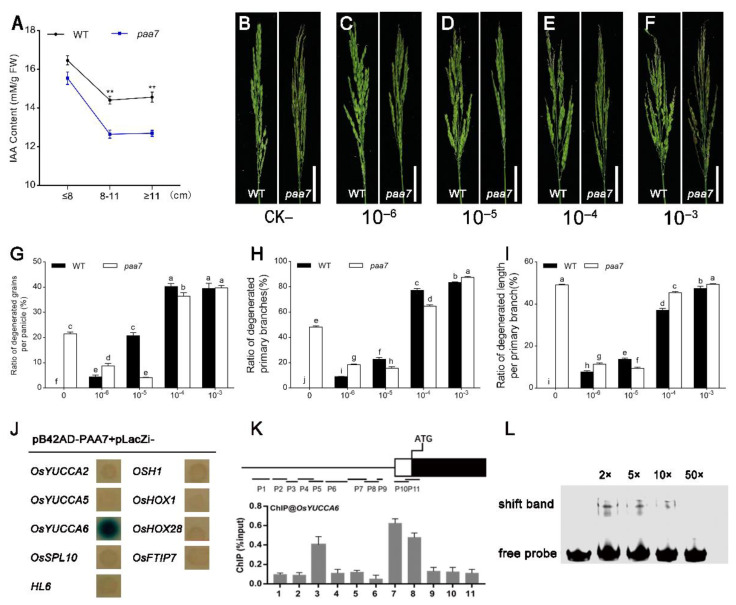
PAA7 regulates apical panicle degeneration through auxin pathway. (**A**) Auxin content in panicle of WT and *paa7* at different developmental stages. Data are shown as means ± SD from three replicates (** *p* < 0.01, Student’s *t*-test). (**B**–**F**) Phenotypes of WT and *paa7* mutant plants treated with different concentrations of auxin. Bars = 5 cm. (**G**–**I**) Ratio of degenerated grains per panicle; ratio of degenerated primary branches; ratio of degenerated length per primary branch (data are shown as means ± SD from 10 replicates. Bars annotated with different letters represent values that are significantly different (*p* < 0.05) according to a one-way ANOVA). (**J**) Y1H assay for interactions between FZP and auxin related genes. (**K**) ChIP-qPCR results show that PAA7 binds to the promoter of *OsYUCCA6.* ChIP enrichment compared with the input sample was tested using qPCR. Data are shown as means ± SD (*n* = 3). (**L**) EMSA assays reveal that PAA7 directly binds to *OsYUCCA6* promoter DNA fragments containing “CCGTTG”.

**Figure 6 ijms-23-09487-f006:**
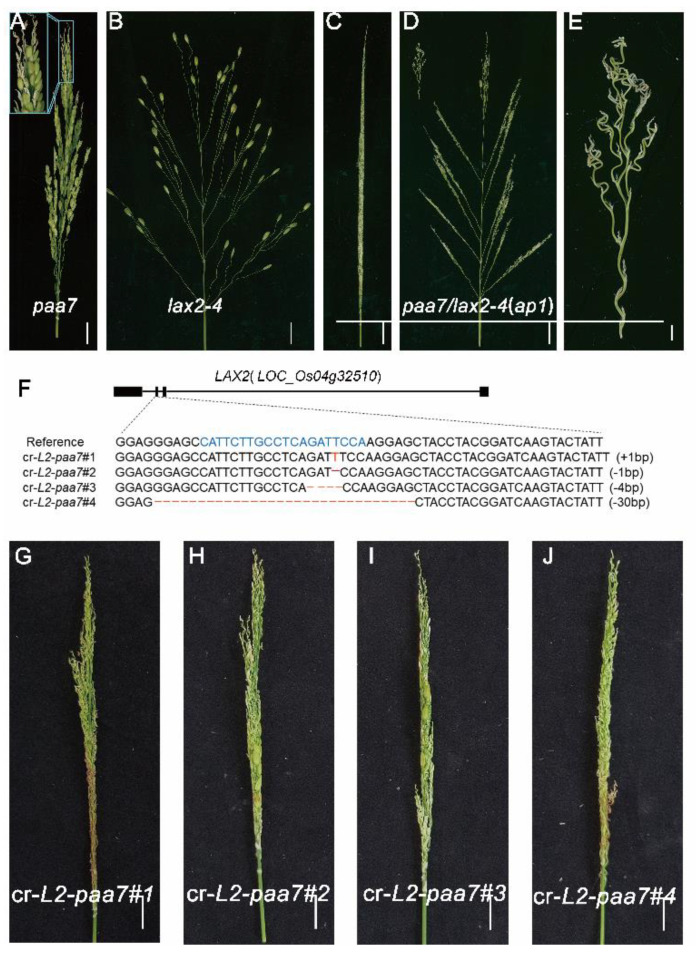
Phenotypic characterization of *paa7/lax2*−*4* double mutants. Panicle of *paa7* (**A**)*, lax2-4* (**B**), *paa7/lax2*−*4* (**C**−**E**), bars = 2 cm. (**F**) Sequencing results of targeted regions of in cr−*L2*−*paa7#1*−*#4* tran−genic plants. Panicle of *cr*−*L2*−*paa7#1* (**G**), *cr*−*L2*−*paa7#2* (**H**)*, cr*−*L2*−*paa7#3* (**I**), and *cr*−*L2*−*paa7#4* (**J**), bars = 2 cm.

**Figure 7 ijms-23-09487-f007:**
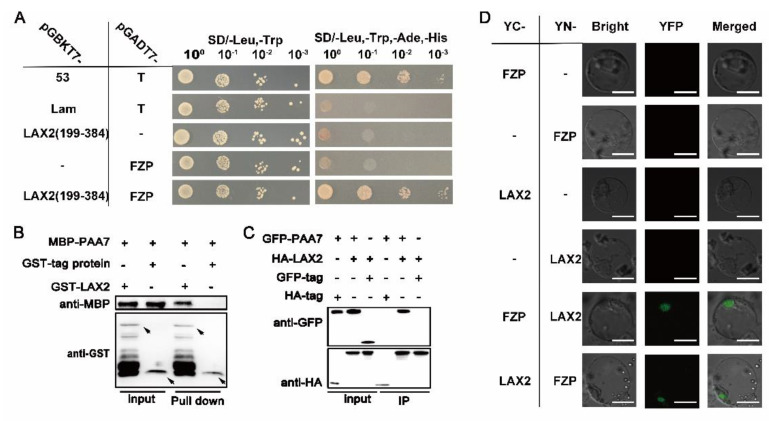
PAA7 interacts with LAX2. (**A**) Yeast two−hybrid assays showed the interaction between PAA7 and LAX2 proteins; pGBKT7−53 and pGADT7−T pair was used as a positive control, the pGBKT7−Lam and pGADT7−T pair was used as negative control, and the pGBKT7−LAX2 and pGADT7 pair was the self-activation test. (**B**) PAA7 binds LAX2 in vitro. MBP−PAA7 were incubated with GST−LAX2 and pulled down by GST−LAX2 and detected by immunoblot with anti−MBP ant-body. (**C**) Co−IP assay confirmation of PAA7 and LAX2 interaction. (**D**) Bimolecular fluorecence complementation (BiFC) assay showed that PAA7 and LAX2 proteins interact in the nucleus of rice protoplasts. Bar = 10 μm.

**Figure 8 ijms-23-09487-f008:**
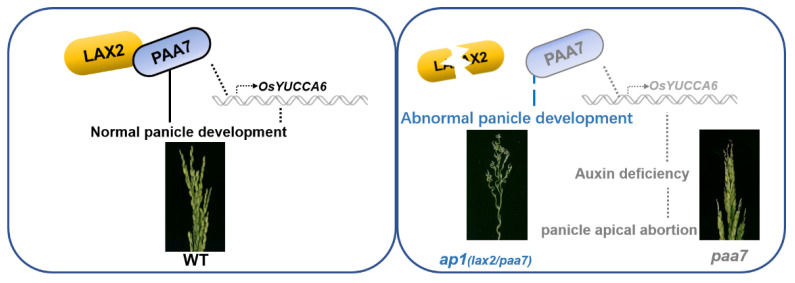
A working model for the FZP in the regulation of panicle development. In WT, PAA7/FZP could interact with LAX2 and activates *OsYUCCA6* to control the development of rice panicle. In the *paa7* mutant, lower expression led to downregulation of *OsYUCCA6* and auxin deficiency in top spikelets of young panicle, finally resulting in panicle apical abortion; the loss of function of LAX2 aggravates the phenotype of *paa7* mutant, which presented as *ap1* (fizzy panicle).

## Data Availability

Not applicable.

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
