# Peer review of "Panicle Apical Abortion 7 Regulates Panicle Development in Rice (Oryza sativa L.)"

_ijms, 2022, doi:10.3390/ijms23169487_

Round 1
Reviewer 1 Report
In their manuscript, Dai et al. characterize a mutant, paa7, which presents several aberrant phenotypes, such as the degeneration of the apical spikelets of the panicle and the consequent reduction of yield of the plants. This phenotype is caused by an increase in cell death. Indeed, paa7 spikelets accumulate more ROS, causing DNA damage.
After a complementation test, it was possible for the authors to hypothesize that PAA7 is a new allele of FZP. Furthermore, PAA7 is able to interact with LAX2.
I have a few suggestions:
· In line 94 something is missing. Indeed, it is not stated the length of what per primary branches were degenerated. Also, in lines 93-95 it is difficult to understand the phenotype that is described: what does it mean that the length of the primary branches is degenerated? Are the primary branches shorter than the wild type? And all the primary branches or just the ones at the top of the panicle?
· I have some suggestions regarding the pictures. Indeed, the presented pictures are not so much informative of the phenotype. For example, in Figure 1C I suggest spreading a bit more the panicle branches to better separate them. This will help in showing clearly the phenotype described. Figure 1J is not clear: it is impossible to see what the white arrow is pointing at. Maybe it will be better to enlarge the part of the panicle to which the authors are referring. In figure 1K I suggest indicating for each wild-type spikelet the length of the panicle from which they were collected, to better understand the progression of development. Also for figures 4 and 5, I suggest taking more explicative pictures of the panicle, to see clearly the described phenotype;
· In the paragraph “PAA7 synergistically work with LAX2 during panicle development” I haven’t really understood why LAX2 is taken into exam. Indeed, I can’t find citation number 27 in English.
· In the discussion, it is not further mentioned the ROS accumulation that leads to cell death instead more experiments regarding auxin signalling are present. The results of such experiments are only presented as supplementary material. My suggestion is to include the part relative to auxin in the results and to better describe and speculate on how auxin deficiency leads to ROS accumulation and cell death in the paa7 mutant.
The discussion is mainly referring to results which are not presented in the main text but in the supplementary material and figures. I would encourage the authors to present those findings they think are important, in the correct part of the manuscript.
Author Response
Response to Reviewer 1 Comments
Point 1: In line 94 something is missing. Indeed, it is not stated the length of what per primary branches were degenerated. Also, in lines 93-95 it is difficult to understand the phenotype that is described: what does it mean that the length of the primary branches is degenerated? Are the primary branches shorter than the wild type? And all the primary branches or just the ones at the top of the panicle?
Response 1: We revised the description of the top degradation trait in the paper (lines 94-96). Degenerated primary branches and spikelets mainly located at the top of panicle became bleached and stopped growing and resulted in shorter panicles in paa7 mutant.
Point 2: I have some suggestions regarding the pictures. Indeed, the presented pictures are not so much informative of the phenotype. For example, in Figure 1C I suggest spreading a bit more the panicle branches to better separate them. This will help in showing clearly the phenotype described. Figure 1J is not clear: it is impossible to see what the white arrow is pointing at. Maybe it will be better to enlarge the part of the panicle to which the authors are referring. In figure 1K I suggest indicating for each wild-type spikelet the length of the panicle from which they were collected, to better understand the progression of development. Also, for figures 4 and 5, I suggest taking more explicative pictures of the panicle, to see clearly the described phenotype;
Response 2:
We have shown all primary branches from top to bottom and the distribution of degenerated grain number in Figure 1D-E, represented the expansion phenotype of paa7 panicle in Figure 1C. The top part of the panicles indicated by the arrow has been enlarged in the blue box in Figure 1J; The length of panicles which spikelets collected from was indicated in stages; images which showed phenotype more clearly were added in figures 4 and 5 and Figure S4.
Point 3: In the paragraph “PAA7 synergistically work with LAX2 during panicle development” I haven’t really understood why LAX2 is taken into exam. Indeed, I can’t find citation number 27 in English.;
Response 3:
As both paa7 and lax2-4 mutants were derived from natural panicle mutation ap1, which shows frizzy panicle phenotype, and the reciprocal crosses of lax2-4 and paa7 showed that F1 progenies showed a wild-type panicle phenotype, we speculated that LAX2-4 and PAA7 may synergistically regulate rice panicle development. Thus, LAX2 is taken into exam. Citation number 27 is available at the following website: https://link.springer.com/article/10.1007/s00344-021-10450-y.
Point 4: In the discussion, it is not further mentioned the ROS accumulation that leads to cell death instead more experiments regarding auxin signalling are present. The results of such experiments are only presented as supplementary material. My suggestion is to include the part relative to auxin in the results and to better describe and speculate on how auxin deficiency leads to ROS accumulation and cell death in the paa7 mutant.
Response 4: We have moved the part related to auxin into “Results” and the related discussion was added.
Point 5: The discussion is mainly referring to results which are not presented in the main text but in the supplementary material and figures. I would encourage the authors to present those findings they think are important, in the correct part of the manuscript.
Response 5: We have moved the part related to auxin into “Results”, and further discussed the auxin related results.
Reviewer 2 Report
This is a well-designed study with all necessary data to draw the conclusion. Presented data showed that PAA7 interacts with LAX2 to regulate the young panicle and spikelet development in rice via auxin signaling pathway, likely by activating OsYUCCA6 expression for auxin biosynthesis. Reciprocal crossings and CRISPR/Cas9-mediated LAX2 targeted mutagenesis approaches provided consistent data. ROS accumulation was detected in the aborted spikelets and the exogenous application of low-level auxin alleviated the mutant phenotypes, further suggesting that disrupted auxin biosynthesis might be the cause of the panicle abortion in the paa7 mutants. Included evidence is solid for the conclusion and new approaches could be developed to increase rice grain yield by engineering paa7 or its regulatory pathway.
There were small minor errors that need to be fixed:
Page 2, line 71: “rice floral” should be “rice floral organs”
Page 2, line 89: “remarkable” should be “remarkably”
Page 2, line 93: “speculate” should be “speculation”
Page 3, line 104: “Except aborted spikelets” should be “In addition to the aborted spikelets”
Figure 1 legend: add the number of samples included in the t-test.
Figure 2 and 3 legends: Add description for the error bars and statistical analysis.
Page 6, lines 203-204: “separation ratio” should be “segregation ratio”
Figure 4I: Add description of the error bars and statistical analysis. What was the sample size?
Author Response
Response to Reviewer 2 Comments
Point 1: In Page 2, line 71: “rice floral” should be “rice floral organs” ;Page 2, line 89: “remarkable” should be “remarkably” ;Page 2, line 93: “speculate” should be “speculation” ;Page 3, line 104: “Except aborted spikelets” should be “In addition to the aborted spikelets” ; Page 6, lines 203-204: “separation ratio” should be “segregation ratio”.
Response 1: We fixed the description error mentioned above.
Point 2: Figure 1 legend: add the number of samples included in the t-test; Figure 2 and 3 legends: Add description for the error bars and statistical analysis; Figure 4I: Add description of the error bars and statistical analysis.
Response 2: We have completed the description of statistical analysis in all legends in this text.
Point 3: Figure 4I: What was the sample size?
Response 3: The sample size was 7-8cm length of panicles and the related information has been added in line 190, page 6
Round 2
Reviewer 1 Report
The new version of the manuscript has been improved according to the suggestions and in my opinion, is now well organized and clearer than the previous one.
I've still some points that should be revised and maybe modified:
- Some phrases need to be adjusted because the language is not correct, as examples (but not exhaustive) I found phrases on these lines containing errors: 75, 223-224, 320-321, 326-327, 331-335, 342-344, 368.
In the result part:
-line 228 change (Figure S3) to (Figure S2)
-line 254: "The same derivation of paa7 (Figure 5A6A) and previously reported lax2-4 (Figure 254 5B6B) mutant [27] indicated that LAX2-4 and PAA7 may synergistically regulate rice panicle development." It is not clear which derivation, please explain better the point.
-line 274 change Figure S4A-D) to Figure S3A-D),
In the discussion part:
It is not clear to me why the authors concluded that LAX2-4 may act upstream of PAA7/FZP. Indeed they present evidence for a direct interaction between LAX2 and FZP as well as genetic interaction, suggesting that the two proteins can cooperate to trigger the correct development of inflorescence architecture and spikelets development. No evidence are presented in this work on LAX2 and the degeneration of the spikelets. The model presented in figure 8 is not supported by the data presented, I cannot understand why the authors at the end propose that LAX2 is upstream to FZP. It would be more appropriate to hypothesize a parallel pathway in which synergetically both FZP and LAX2 control panicle and spikelets development.
Round 3
Reviewer 1 Report
The new version of the manuscript is well organized and clearly presented. I was wondering if it could be possible to speculate an effect of paa7 mutation which can define a new weak allele of FZP, which when combined with lax2-4 result in a complete loss of the function of both genes.